# Predicting the Coping Skills of Older Drivers in the Face of Unexpected Situation

**DOI:** 10.3390/s21062099

**Published:** 2021-03-17

**Authors:** Yusuke Kajiwara, Haruhiko Kimura

**Affiliations:** Department of Production Systems Engineering and Sciences, Komatsu University, Shichomachi Nu1-3, Komatsu, Ishikawa 923-8511, Japan; haruhiko.kimura@komatsu-u.ac.jp

**Keywords:** prediction of coping skills, unexpected situation, older driver

## Abstract

In recent years, when an older driver who cannot immediately recognize, judge, and operate properly faces an unexpected situation, they often panic, which may cause a traffic accident. However, there has not yet been enough discussion about the coping skills of older drivers in the face of this unexpected situation. Therefore, this study discusses the coping skills of older drivers in the face of unexpected situations. Moreover, we propose a coping skills prediction system (CP system). The CP system predicts coping skills from the tilt angle and angular velocity of the left foot when an older driver is driving or preparing to start a car. The experiment carried out two phases, a phase of driving a car and a phase of preparing to start the car. In the driving phase, the young and older driver drive the car in a driving simulator. The average age of the young driver group was ± standard deviation = 20.6 ± 0.7 years, and the age of the older driver group was 78.5 ± 5.1 years. The driving route included 15 cases in which collision accidents are likely to occur. We analyzed the experimental results of the driving phase and clarified the predictors of coping skills. Moreover, we analyzed the correlation between the left foot movement in driving and the left foot movement during preparing to start the car. As a result of the experiment, there was a 0.84 correlation between the tilt angle of the left foot of the older driver in driving and the tilt angle of the left foot of the older driver in preparing to start the car. The result shows that the coping skills can be predicted from the tilt angle of the left foot of the older driver during preparing to start the car. We showed that the coping skill can be predicted with an accuracy of 92% or 94% on average from the tilt angle and the angular velocity of the left foot while driving or preparing to start the car. Moreover, we clarified that the tilt angle of the left foot of a driver without coping skills is perpendicular to the ground compared to a driver with coping skills. This study is expected to contribute to the prevention of traffic accidents that occur in the face of an unexpected situation.

## 1. Introduction

In recent years, when an older driver who cannot immediately recognize, judge, and operate properly faces an unexpected situation, they often panic, which may cause a traffic accident [1]. The problem is addressed by suspending the driver’s license for drivers aged 75 and over who have diminished driving skills. At present, older drivers measure their judgment and cognitive ability through questionnaires such as the Mini-Mental State Examination (MMSE) [2] and the Hasegawa dementia scale-revised (HDS-R) [3]. When the doctor determines that the older driver has dementia, the driver’s license is suspended. Besides, in Japan, the Road Traffic Law was amended, and by 2022, a driving skills test will be obligatory for drivers aged 75 and over who have a history of a violation. Therefore, there is a need to develop a system for objectively measuring driving skills.

To support driving tasks and improve driving safety, related works reported the effects of older driver characteristics and driver status (behavior, attention, intention, drowsiness, malaise, cognitive ability, and mental state) on driving behavior [4,5,6]. Older drivers respond slower than younger drivers due to aging [7]. Older drivers have diminished perceptual and cognitive ability such as visual attention [8], visual processing speed [9], and working memory [10]. Therefore, older drivers are slower to respond to unexpected situations than younger drivers. When an older driver feels uneasy about driving due to this decrease in perceptual–cognitive ability, the older drivers drive improperly, such as making sudden lane changes [11,12]. These facts suggest that older drivers need to be prepared in advance to deal with unforeseen situations. However, there is still not enough discussion about the coping skills of older drivers in the face of unexpected situations.

This study discusses the coping skills of older drivers in the face of unexpected situations. We experimented and obtained not only the movement of the driver’s left foot, but also the amount of operation of the brake, accelerator, and steering wheel, autonomic nerve activity, and the amount of brain activity. The experiment carried out two phases, a phase of driving a car and a phase of preparing to start the car. In the driving phase, the young and older driver drove the car in a driving simulator. The average age of the young driver group was ± standard deviation = 20.6 ± 0.7 years, and the age of the older driver group was 78.5 ± 5.1 years. The driving route included 15 cases in which collision accidents are likely to occur. We analyzed the experimental results of the driving phase and clarified the predictors of coping skills. Moreover, we analyzed the correlation between the left foot movement in driving and the left foot movement during preparing to start the car. As a result, there was a significant difference in braking and steering operation. This result is similar to the result of related works [13,14]. Related works [14] focus on these indicators as new measures of driving performance. In addition to this result, as a new finding, we clarified that the tilt angle of the left foot of a driver without coping skills is perpendicular to the ground, compared to a driver with coping skills. The results of this analysis suggest that the movement of the older driver’s left foot is a predictor of collision accidents for the older driver. There was a 0.84 correlation between the tilt angle of the left foot of the older driver in driving and the tilt angle of the left foot of the older driver in preparing to start the car. The result shows that the coping skills can be predicted from the tilt angle of the left foot of the older driver during preparing to start the car. We showed that the coping skills were predicted with an accuracy of 92% or 94% on average from the tilt angle and the angular velocity of the left foot while driving or preparing to start the car. This study is expected to contribute to the prevention of traffic accidents that occur in the face of an unexpected situation.

## 2. Development of Coping Skills Prediction System

### 2.1. Definition of Coping Skills

The older drivers are prone to dynamic hazard accidents such as collisions with other vehicles [15]. The driver’s physical function is impaired by mental stress [16]. We define coping skills as skills to properly step on the brake and stop the car when faced with an unexpected dynamic hazard under a mental workload such as impatience or nervousness. A driver with coping skills is defined as “CS”. A driver without coping skills is defined as “NCS”. The Venn diagram of CS and NCS is shown in Figure 1.

Some drivers have poor driving skills and cognitive abilities. Under the current driver’s license renewal process, the driver’s license will be suspended or revoked for drivers with poor driving skills and cognitive abilities. The driver’s license renewal system prevents accidents caused by drivers with poor driving skills and cognitive abilities. On the other hand, the current driver’s license renewal system measures driving skills and cognitive abilities during normal times and decides to renew the driver’s license. Therefore, a driver who has renewed his driver’s license may not be able to deal with unexpected situations. Therefore, we will further subdivide the driver who was able to renew the driver’s license into NCS and CS. The coping skills prediction (CP) system prevents accidents caused by NCS by encouraging NCS to return the driver’s license.

### 2.2. Fundamental Principle

In this study, we propose a coping skills prediction system (CP system). Figure 2 shows an outline of the CP system.

The CP system predicts coping skills from the tilt angle and angular velocity of the left foot when an older driver is driving or preparing to start a car. Machine learning is used for prediction. The driver’s movements when driving a car are different from the driver’s movements when preparing to start the car. However, these movements have in common that the driver recognizes, judges, and controls the movement within the time limit. Therefore, a correlation is expected between the driver’s movements during preparation for starting the car and the driver’s movements during driving the car. Generally, when the angle of tilt of the left foot is perpendicular to the ground, the driver cannot sufficiently stabilize a body while the driver drives a car. Since the muscle strength of older drivers is weakened [17], it is expected that the tilt angle of the left foot has a large effect on the stability of the body. Therefore, when the tilt angle of the left foot of an older driver is perpendicular to the ground, it may not be possible to deal with unexpected situations immediately.

The driver feels stressed when a driver faces an unexpected situation. Mental stress induces muscular tension [16]. Muscle tension reduces the range of motion of joints. When the driver steps on the brake with his right foot, the driver moves his right foot forward to the left because the brake is next to the accelerator. Therefore, when the driver feels stressed and the range of motion of the joints is reduced, the driver may not be able to brake properly.

The amount of braking can be obtained directly from the car control device. The amount of braking can be obtained indirectly from the movement of the right foot. However, it is difficult to measure the stability of the body and muscle tension. Therefore, in this study, we focused on the tilt angle and angular velocity of the left foot. The tilt angle of the left foot and the range of motion of the joint are acquired by the tilt measurement unit (IMU) attached to the left foot. The range of motion of a joint is expressed by the angular velocity of the IMU. The CP system inputs the tilt angle and angular velocity of the left foot into machine learning to predict coping skills.

### 2.3. Acquisition of Tilt Angle and Angular Velocity of the Left Foot by IMU

We used ATR-Promotion’s compact wireless multifunction sensor “TSND151” to acquire the movement of the driver’s left foot. Figure 3 shows the coordinate system of TSND151. The coordinate system of TSND151 is the right-handed coordinate system.

The TSND151 is equipped with MPU-9250, which measures the acceleration, angular velocity, and geomagnetism of each axis. The measurement range of acceleration is ± 16 G, the resolution is 16 bits, and the sensitivity is 0.5 mG. The measurement range of angular velocity is ±2000 deg/sec, the resolution is 16 bits, and the sensitivity is 0.06 deg/sec. The measurement range of geomagnetic is ±4800 μT, the resolution is 16 bits, and the sensitivity is 0.15 μT/LSB. These measurements are sent to the information terminal via Bluetooth 2.0. The size of the TSND151 is 40 mm (W) × 50 mm (H) × 14 mm (D). The weight of the TSND151 is 27 g. The TSND151 is a small and lightweight IMU. Therefore, even if the TSND151 is attached to the driver’s left foot, the driver can operate as usual. The tilt angle of TSND151 is calculated from the acceleration sensor, angular velocity sensor, and geomagnetic sensor. The IMU is worn on the driver’s left ankle, using a wristband. The knee is in the positive direction of the IMU’s *x*-axis. The tilt angle on the *y*-axis represents the tilt angle of the left foot. A tilt angle of −90 degrees on the *y*-axis means that the tilt angle of the left foot is perpendicular to the ground. Angular velocity on the *x*-axis represents the range of motion of joints. The performance of the IMU is accurate enough to measure the tilt angle of the left foot and the range of motion of the joints of an older driver.

### 2.4. Variable Selection by a Hypothesis Test

Effective explanatory variables were selected to improve the accuracy of machine learning. Explanatory variables are generally selected by principal component analysis, information criteria such as Akaike’s Information Criterion (AIC), and hypothesis test. In this study, explanatory variables were selected by hypothesis test. The Brunner–Munzel test [18] selects variables that are effective in predicting CS and NCS. The Brunner–Munzel test is a non-parametric test. The Brunner–Munzel test has high statistical power, even in small samples. The characteristic elements are the average, variance, maximum value, and minimum value of the acceleration, angular velocity, and tilt angle of each axis. We made a null hypothesis that there is no difference in a characteristic element of both groups of a CS and NCS and conducted the Brunner–Munzel test. The rejection region is 0.01. When the null hypothesis is rejected, a characteristic element has a significant difference between the two groups. Therefore, the characteristic elements selected by the hypothesis test are effective variables to predict CS and NCS. Explanatory variables of machine learning are consisting of the characteristic elements selected by the Brunner–Munzel test.

### 2.5. Prediction of Coping Skills by Machine Learning

We constructed a prediction model using machine learning. Machine learning makes it is easier to construct models than the classic and robust approach [19,20]. There are various types of machine learning such as support vector machine (SVM) [21], naive Bayes [22], and random forest [23]. In this study, we use SVM, random forest, and naive Bayes to evaluate the prediction accuracy of coping skills. SVM is a method of learning support vectors and has high performance in binary classification. The naive Bayes classifier is a probabilistic model based on the independence assumption and Bayes’ theorem. Random forest decides the prediction result of the decision tree by the majority and outputs the final prediction result. The decision tree is constructed with the classification and regression tree (CART) [24].

## 3. Verification Experiment

### 3.1. Experimental Design

Experiments were conducted to verify the following:We show effects by predicting CS and NCS.We clarify predictors of coping skills.We show the prediction accuracy of the CP system.

The experiment carried out two phases, a phase of driving a car and a phase of preparing to start the car. The driving phase was conducted to verify the definitions of CS and NCS. We analyzed the experimental results of the driving phase and clarified the predictors of coping skills. In the driving phase, the driver drives the car in a driving simulator. The driver drives a car along a preset driving route. As shown in Table 1, the driving route includes 15 cases in which collision accidents are likely to occur.

The data for 30 s before the time when the driver passed each case was used for the analysis. However, when an accident occurred in a scene other than the 15 cases, that scene was skipped. The subject drove several routes in the driving simulator before the experiment to get used to the operation of the driving simulator. In the driving phase, I and II control experiments were conducted.

I.The driver drives the route including the cases in Table 1 for the first time.II.After conducting Experiment I, the driver redrives the same route as Experiment I.

We instructed the subjects to drive the route within 10 min in Experiment I. In Experiment I, the driver drives the route containing the scenes in Table 1 for the first time. The driving route contains 15 cases that are collision accident-prone. In Experiment I, the 15 cases in Table 1 are unexpected situations since the driver drives on an unfamiliar road. In Experiment I, the driver feels impatient and nervous because the driver must drive the route within the time limit. Therefore, by conducting Experiment I, the coping skills of the subjects can be measured. Subjects with a small number of accidents in Experiment I have high coping skills. Since Experiment II redrives the same route as Experiment I, the subject knows in advance the scene where a collision accident is likely to occur. Therefore, II can measure normal driving skills. The average number of accidents in Experiment II indicates the driving skills of a general driver when driving on a familiar route. Therefore, by comparing the number of accidents in Experiment I and II, it is likely to find subjects who cannot deal with unexpected situations.

Control experiments of Experiment III and IV were conducted to show that CS and NCS can be predicted from the movement of the left foot when preparing to start the car.

III.The driver prepares to start the car within the time limit.IV.The driver prepares to start the car.

The movement when preparing to start the car was defined as the movement when the subject fastened the seatbelt and stepped on the brake. Subjects performed Experiment III and Experiment IV 10 times each. We analyzed the correlation between the left foot movement in driving (Experiment I and II) and the left foot movement during preparing to start the car (Experiment I and II). By analyzing the correlation, we show that the coping skills can be predicted from the movements of the left foot of the older driver while preparing to start the car.

### 3.2. Data Acquisition

The measurement environment is shown in Figure 4
Table 2 shows the specifications of the measuring equipment. In this experiment, the movements of the left foot of the subjects and the amount of braking, accelerator, and steering operations were measured to acquire behavioral characteristics for dealing with unexpected situations. Since the amount of brake, accelerator, and steering is acquired, it is not necessary to measure the movements of both hands and right foot related to these operations. The amount of autonomic nerve activity is measured to objectively acquire the degree of nervousness. When the sympathetic nerve is predominant, it indicates that the subject gets nervous. Frontal lobe brain activity was measured to capture subject cognitive activity. The movement of the subject while driving or preparing to start the car was acquired by the IMUs. IMUs were attached to the left foot of the subject. The IMU measures the acceleration, angular velocity, and tilt angle of each axis. The amount of brain activity in the frontal lobe of the subject is acquired by near-infrared spectroscopy (NIRS). NIRS was worn on the subject’s head. NIRS measures frontal cerebral blood flow on 22 channels. The frontal lobe is the part of the brain that controls cognitive skills. The driver’s frontal cerebral blood flow represents the cognitive load on the driver. Therefore, NIRS can directly measure cognitive load. A related study uses the Detection Response Task (DRT) [25]. DRT indirectly measures cognitive load by asking the driver to respond when they detect a small light in their peripheral vision. DRT is an international standard, which can be safely applied with no appreciable effect on driving performance. However, in this study, we did not use DRT because we used NIRS to measure cognitive load directly. The amount of autonomic nerve activity is acquired by a pulse wavemeter. The pulse wave sensor measures very low frequency (VLF), low frequency (LF), high frequency (HF), total power (TP), heart rate (HR), and R–R interval (RR). As the characteristic elements of this experiment, the average, variance, maximum value, and minimum value of the measured values were calculated.

In this experiment, the subject’s subjective degree of impatience was measured by the subject’s self-report. The subjects were asked to verbally answer to their subjective impatience when encountering each case in Experiment I and Experiment II. The subjects were asked to verbally answer the subjective degree of impatience after preparing to start the car. The degree of impatience was measured by the 7-point scale. In Experiment Is and II, the number of accidents was recorded.

### 3.3. Subjects

The subjects were 13 young drivers and 14 older drivers. However, the CP system cannot be applied to drivers who step on the accelerator with their right foot and brake with their left foot due to the principle of prediction in the CP system. There were two young drivers and two older drivers who used both feet to step on the accelerator and brakes. Four older drivers declined the experiment because of physical disabilities. The average age of the young driver group was ± standard deviation = 20.6 ± 0.7 years, and the age of the older driver group was 78.5 ± 5.1 years. The oldest driver in the elderly driver group was 91 years old. A cognitive ability test [3] was performed on subjects in the older driver group. Older drivers were classified as Group C, Group B, and Group A according to their cognitive ability test scores. Older drivers in group C have low memory and judgment. Older drivers in Group B have slightly lower memory and judgment. Older drivers in Group A have no problems with memory and judgment. Older drivers were classified into group C with a cognitive ability test score of less than 49, group B with a score of 49 or more and less than 76, and group A with a score of 76 or more. There were two older drivers in Group B, but the scores were 69.8 points and 74.9 points, which were near the borderline between Group A and Group B. The other older drivers were in Group A.

## 4. Results

### 4.1. Categorizing Subjects into CS and NCS

We analyzed the subjective impatience and the difference in the number of accidents between the young driver group and the older driver group using the Brunner–Munzel test. The null hypothesis is that there is no difference between the young driver group and the older driver group in terms of subjective impatience and the number of accidents in Experiment I and Experiment II. The rejection region is 1%. The results of the test are shown in Table 3. The difference between the two groups in terms of subjective impatience and the number of accidents was not statistically significant.

Next, we analyzed the difference in the subjective impatience and the number of accidents between Experiment I and Experiment II using the Brunner–Munzel test. The set of young drivers and older drivers is defined as ∑. The null hypothesis is that there is no difference between Experiment I and Experiment II in terms of subjective impatience and the number of accidents of ∑. The rejection region is 1%. The results of the test are shown in Table 4. The difference between the two groups in terms of the number of accidents was statistically significant. The test results show that the number of accidents in Experiment I is larger than that in Experiment II. The difference between the two groups in terms of subjective impatience was not statistically significant. However, the subjects were likely more nervous in Experiment I than in Experiment II, even though there was no significant difference in subjective impatience. The null hypothesis is that there is no difference between Experiment I and Experiment II in terms of the pulse wave sensor value ∑. The rejection region is 1%. Table 5 shows three characteristic elements with high statistics.

The difference between the two groups in terms of average, minimum, and maximum HR was statistically significant. When the subject is nervous, the subject’s sympathetic nerves dominate. When the sympathetic nerve is dominant, the HR increases.

The difference between the control conditions of Experiment I and Experiment II is whether an unexpected situation occurs while the driver is driving. Besides, the subjects were more nervous in Experiment I than in Experiment II. Therefore, these two factors may have influenced the number of accidents in Experiment I. Therefore, it is likely that these two factors affected the number of accidents in Experiment I. NCS is a driver who cannot deal with the unexpected situation of Experiment I. NCS is expected to have more accidents in Experiment I than in Experiment II. Therefore, drivers are categorized into CS and NCS based on the number of accidents in Experiment I.

The average ± standard deviation of the number of accidents in Experiment II was 1.6 ± 1.1. Therefore, subjects with more than two accidents in Experiment I were categorized into NCS. The other subjects were categorized into CS. Eight subjects were categorized into NCS in the young driver group, and the other subjects in the young driver group were categorized into CS. Five subjects in the elderly driver group were categorized into NCS, and the other subjects in the older driver group were categorized into CS.

We analyzed the difference in the subjective impatience and the number of accidents between the NCS group and CS group using the Brunner–Munzel test. The null hypothesis is that there is no difference between the NCS group and the CS group in terms of subjective impatience and the number of accidents in Experiment I and Experiment II. The rejection region is 1%. The results of the test are shown in Table 6.

Since the threshold was set based on the number of accidents in Experiment I and the CS group and NCS group were categorized, the number of accidents in Experiment I could not be tested. However, from Table 6, there is a definite difference in the number of accidents in Experiment I between the CS group and the NCS group. Drivers in the NCS group have more than twice as many accidents as drivers in the CS group. The difference between the two groups in terms of subjective impatience and the number of accidents was not statistically significant.

The average ± standard deviation of Experiment III was 3.8 ± 1.5, and the average ± standard deviation of Experiment IV was 2.3 ± 1.5. The degree of impatience of Experiment III and IV was less than 4, and the subjects did not feel impatience.

### 4.2. Behavioral and Cognitive Characteristics of CS and NCS

Table 6 shows that drivers in the NCS group are more likely to have an accident when faced with an unexpected situation under a mental workload such as impatience or nervousness. To investigate the effect of impatience on behavior and cognition, the characteristics of behavior and cognition when the driver feels impatience or calm were analyzed using the Brunner–Munzel test. The null hypothesis is that there is no difference between the NCS group and CS group in terms of the characteristics of behavior and cognition when the driver feels impatience or calm. The rejection region is 1%. Table 7 and Table 8 show the behavioral and cognitive characteristics of the older driver when they feel impatient or calm. Table 9 and Table 10 show the behavioral and cognitive characteristics of the young driver when they feel impatient or calm.

Impatience is defined as a mental state when subjective impatience is greater than 4. Calmness is defined as a mental state in which subjective impatience is less than 4. According to related work, when an older driver feels uneasy about driving, the older drivers drive at a speed well below the speed limit [11]. A related study reported that steering speed is a good indicator for detecting that a driver has urgently avoided a collision [13]. Related works focus on the maximum of braking as new measures of driving performance [14]. Therefore, the average of accelerator and steering, and the maximum value of braking were analyzed. We analyze the average of the tilt angle in the *y*-axis and the angular velocity in the *x*-axis, which are necessary elements for the principle of the CP system. We analyzed good indicators in the characteristic of cognitive to detect older groups of CS and NCS.

The characteristics of the behavior in Table 7, Table 8, Table 9 and Table 10 are the average accelerator and steering, the maximum of braking, tilt angle in the *y*-axis, and the average angular velocity in the *x*-axis. The characteristics of the cognitive in Table 7, Table 8, Table 9 and Table 10 are the average of the measurements of each channel of NIRS. Table 7, Table 8, Table 9 and Table 10 show the average of cerebral blood flow in channel 6 and channel 18 with the highest statistics when older drivers were feeling impatient. Figure 5 and Figure 6 show heat maps of brain activity. The heat map of brain activity was normalized with the maximum values in Figure 5 and Figure 6 as 1. Scatter plots of the characteristics of behavior and cognition are shown in Figure 7 and Figure 8.

*θ*^(*y*)^_avg_ and ω^(*x*)^_avg_ in Figure 7 are defined as the average of *y*-axis tilt angle and angular velocity in the *x*-axis. *C*^(6)^_avg_ and *C*^(18)^_avg_ in Figure 8 are defined as the average of cerebral blood flow in channel 6 and channel 18. From Table 7, the difference between the two groups in terms of the maximum of braking, *θ*^(*y*)^_avg_, ω^(*x*)^_avg_, *C*^(6)^_avg_, and *C*^(18)^_avg_ of the older driver who feels impatience were statistically significant. From Table 8, the difference between the two groups in terms of *θ*^(*y*)^_avg_, *C*^(6)^_avg_, and *C*^(18)^_avg_ of the older driver who felt calm were statistically significant. The older drivers in the CS group who felt impatient stepped on the brake more strongly than the older drivers in the NCS group. When the older driver felt calm, the difference between the two groups in terms of the maximum of braking was not statistically significant. When the older driver felt impatient or calm, the tilt angle of the left foot of the older driver in the NCS group was more perpendicular to the ground than that in the CS group. When older drivers felt impatient, older drivers in the CS group had a higher ω^(*x*)^_avg_ than older drivers in the NCS group. The difference between the two groups in terms of ω^(*x*)^_avg_ of the older driver who felt calm was not statistically significant. From Table 9, the difference between the two groups in terms of the average of the accelerator and *θ*^(*y*)^_avg_ of a young driver who feels impatience were statistically significant. From Table 10, the difference between the two groups in terms of the average of *θ*^(*y*)^_avg_, ω^(*x*)^_avg_, *C*^(6)^_avg_, and *C*^(18)^_avg_ of a young driver who feels calm were statistically significant. From Table 9, the young drivers in the NCS group who felt impatient stepped on the accelerator more strongly than the young drivers in the CS group. When the young driver felt calm, the difference between the two groups in terms of the average accelerator was not statistically significant. When the young driver felt impatient or calm, the tilt angle of the left foot of the young driver in the CS group was more perpendicular to the ground than that in the NCS group. The results show that behavioral characteristics differ between younger and older drivers. When young drivers felt calm, young drivers in the CS group had a higher ω^(*x*)^_avg_ than young drivers in the NCS group. When the young driver and older driver felt impatience or calm, the drivers in the CS group had higher cerebral blood flow than drivers in the NCS group.

The characteristics of behavior and cognition in Experiment III and IV were analyzed using the Brunner–Munzel test. The null hypothesis is that there is no difference between the NCS group and the CS group in terms of the behavioral characteristics in Experiment III and IV. The rejection region is 0.01. Table 11 and Table 12 show the behavioral characteristics of the NCS group and CS group in Experiment III and IV. Scatter plots of the characteristic of behavior in Experiment III and IV are shown in Figure 9.

From Table 11, the difference between the two groups in terms of *θ*^(*y*)^_avg_ of the older driver in Experiment III was statistically significant. From Table 12, the difference between the two groups in terms of *θ*^(*y*)^_avg_ of young driver and older driver in Experiment IV was statistically significant.

We analyzed the correlation between the left foot movement in driving and the left foot movement during preparing to start the car using the tests for non-correlation. The null hypothesis is that there is no correlation between driving (Experiment I and II) and preparing to start the car (Experiment III and IV) in terms of the left foot movement. The rejection region is 0.01. Table 13 shows the correlation coefficients between *θ*^(*y*)^_avg_ and ω^(*x*)^_avg_ in driving and the left foot movement *θ*^(*y*)^_avg_ and ω^(*x*)^_avg_ in preparing to start the car.

From Table 13, there was a strong correlation between *θ*^(*y*)^_avg_ of the older driver in driving and *θ*^(*y*)^_avg_ of the older driver in preparing to start the car. Therefore, the coping skills of the older driver can be predicted from *θ*^(*y*)^_avg_ in preparing to start the car.

### 4.3. Prediction of Coping Skills by CP System

We evaluated the prediction accuracy of the CP system. The CP system predicts the coping skills of the older driver from the driving behavior when the older driver feels impatient. The objective variable is coping skills. The labels of coping skills are NCS or CS. The explanatory variables are *θ*^(*y*)^_avg_ and ω^(*x*)^_avg_ of the older driver while driving the car. Radial basis function (RBF)-SVM, naive Bayes, and random forest are used for binary classification. These machine learnings were implemented in the R language. Random forest was built using the random forest package. RBF-SVM and naive Bayes were built using the e1071 package. The parameters of each machine learning are default values in each package. For example, the decision tree of random forest is constructed with the classification and regression tree (CART) [24]. We constructed 500 decision trees. The decision tree is constructed with √D variables. D is the number of dimensions. Decision trees are not pruned. The CP system predicts the coping skills of the older driver from the behavior in preparing to start the car. The objective variable is coping skills. The labels of coping skills are NCS or CS. The explanatory variables are *θ*^(*y*)^_avg_ and ω^(*x*)^_avg_ of the older driver while preparing to start the car. RBF-SVM, naive Bayes, and random forest are used for binary classification. The parameters were the same as the parameters in predicting the coping skills of older drivers based on driving behavior. There are two missing values in the NCS group data of Experiments III and IV due to sensor malfunction. Leave one subject out cross-validation (LOSO-CV) was used for evaluation. Figure 10 shows the predicted results of coping skills using SVM, naive Bayes, and random forest. From Figure 10, the average prediction accuracy of random forest was the highest. Therefore, we focused on the prediction accuracy of random forest. Table 14 shows the results of predicting coping skills of the older driver based on driving behavior using random forest.

The CP system predicted with 92% accuracy on average based on *θ*^(*y*)^_avg_ and ω^(*x*)^_avg_ of the older driver while driving the car. Table 15 shows the results of predicting coping skills of the older driver based on the behavior in preparing to start the car using random forest.

The CP system predicted with 94% accuracy on average based on *θ*^(*y*)^_avg_ and ω^(*x*)^_avg_ of the older driver while preparing to start the car. Therefore, the CP system was able to predict coping skills from the tilt angle and angular velocity of the left foot when an older driver is driving or preparing to start a car.

## 5. Discussion

### 5.1. Expected Effects by Predicting CS and NCS

We discuss the expected effects of predicting CS and NCS. Table 3 and Table 4 show that the number of accidents for both young and older drivers in Experiment I was larger than that in Experiment II. The difference between the control conditions of Experiment I and Experiment II was whether an unexpected situation occurred while the driver was driving. Related works reported that both young and older drivers decrease in performance due to increased roadway complexity [26,27,28]. Related works reported that when drivers encounter complex visual environments or unknown situations, their mental workload increases, and their visual search, danger perception, and vehicle control capabilities become inefficient [29,30]. In Experiment I, the driver drove a car under a heavy mental workload because the driver faced an unexpected situation. From Table 5 the subjects were more nervous in Experiment I than in Experiment II. The drivers who were nervous in the NCS group were more likely to have an accident when faced with an unexpected situation. Therefore, the increase in the number of accidents in I is consistent with the findings of related works. When drivers in the NCS group can be predicted, we can expect that the drivers in the NCS group may reduce the number of accidents by receiving guidance based on the coping skills of drivers in the CS group.

The CP system predicted coping skills to properly step the brake and stop the car when faced with an unexpected dynamic hazard under a mental workload such as impatience or nervousness. DRT indirectly measures cognitive load. The measurement target is different between the CP system and DRT. The CP system has the following advantages:(A)No special equipment is required. The tilt angle of the left foot can be obtained from the in-vehicle camera.(B)The CP system is inexpensive.(C)The CP system can predict coping skills to properly step the brake and stop the car when faced with an unexpected dynamic hazard under a mental workload.

However, it must be kept in mind that the CP system predicts some of the capabilities required to drive. Therefore, it is necessary to comprehensively evaluate the driving skills of the driver by using a method such as DRT and a CP system together.

### 5.2. Behavioral and Cognitive Characteristics of CS and NCS

From Table 7 and Figure 7a,b, the difference between the CS group and NCS group in terms of the *θ*^(*y*)^_avg_ of the older driver who feels impatience or calm were statistically significant. When the older driver felt impatient or calm, the tilt angle of the left foot of the older driver in the NCS group was more perpendicular to the ground than that in the CS group. Therefore, the driver in the NCS group is likely to be unable to stabilize their body sufficiently while driving a car. Since *θ*^(*y*)^_avg_ in the CS group was significantly different *θ*^(*y*)^_avg_ in the NCS group when the driver felt impatient or calm, it is suggested that *θ*^(*y*)^_avg_ is an independent index that is less related to impatience and nervousness. On other hand, from Table 8 and Figure 7b, the difference between the CS group and NCS group in terms of the ω^(*x*)^_avg_ of the older driver who feels impatient was statistically significant. When older drivers felt impatient, the older drivers in the NCS group had a smaller ω^(*x*)^_avg_ than older drivers in the CS group. The older drivers in the NCS group who felt impatient had a smaller ω^(*x*)^_avg_ than older drivers in the NCS group who felt calm. The driver feels stressed when the driver faces an unexpected situation. Mental stress induces muscular tension [16]. Muscle tension reduces the range of motion of joints. When the driver steps on the brake with his right foot, the driver moves his right foot forward to the left because the brake is next to the accelerator. Therefore, the range of motion of the hip joint can be expressed by the ω^(*x*)^_avg_. These results suggest that the hip joint of the older driver was narrowed due to overstrain when the older driver felt impatient. From Table 7, the difference between the CS group and NCS group in terms of the maximum of braking of the older driver who feels impatience were statistically significant. This result suggests that the older drivers in the CS group were able to apply a sudden brake and stop suddenly when faced with an unexpected situation, even when they felt impatient. On other hand, the older driver in the NCS group is likely to be unable to apply the brake properly because the range of motion of the hip joints of the older driver in the NCS group is narrowed when they feel impatience and nervousness. Since the difference between the CS group and NCS group in terms of the ω^(*x*)^_avg_ of the older driver who feels calm were not statistically significant, it is suggested that ω^(*x*)^_avg_ is a dependent index related to impatience and nervousness. From the above results, the CP system can predict coping skills of the older driver from *θ*^(*y*)^_avg_ and ω^(*x*)^_avg_ by judging indirectly the stability of the body while the older driver drives a car and the narrowing of the hip joint due to overstrain.

From Table 7 and Table 8, the difference between the CS group and NCS group in terms of *C*^(6)^_avg_ and *C*^(18)^_avg_ of the older driver who feels impatience or calm were statistically significant. Older drivers in the CS group have higher frontal cerebral blood flow than older drivers in the NCS group. The frontal lobe is the part of the brain that controls cognitive skills. The driver’s frontal cerebral blood flow represents the cognitive load on the driver. Older drivers in the CS group used more cognitive skills than older drivers in the NCS group. Therefore, it was suggested that the older drivers in the CS group were more aware of changes in the surrounding environment than the older drivers in the NCS group; as result, the older drivers in the CS group were activating the frontal lobe. The older drivers between the ages of 65 and 80 are prone to dynamic hazard accidents such as collisions with other vehicles [15]. It was suggested that older drivers in the CS group were aware of changes in their surroundings and were ready to deal with unforeseen circumstances, which led to a reduction in dynamic hazard accidents.

The behavioral characteristics of young drivers were different from those of older drivers; the tilt angle of the left foot of the young driver in the CS group was more perpendicular to the ground than that in the NCS group. Therefore, young drivers were not prepared for their left foot to deal with an unexpected situation. The difference between the CS group and NCS group in terms of the *ω*^(*x*)^_avg_ of a young driver who feels impatience was not statistically significant. Young drivers have more muscle strength than older drivers [17] and can easily stabilize their bodies while driving a car. The young drivers are more responsive than older drivers [7]. Therefore, young drivers can respond immediately, even if they are not prepared to deal with unexpected situations. The cause of a young driver’s accident is different from that of an older driver. The related works point out that young drivers and older drivers have different causes of road accidents [15]. Young drivers between the ages of 15 and 24 generally have a relatively high accident rate of static hazards, such as collisions with fixed objects. Young drivers have lower driving skills and are more vulnerable to risk-taking than adult drivers [31,32]. From Table 9, the difference between the CS group and NCS group in terms of the average of the accelerator of a young driver who feels impatient was statistically significant. The average accelerator of young drivers in the NCS group who are feeling impatient is the highest. Therefore, it was suggested that the young driver of the NCS group who felt impatient drove a car at a speed exceeding the speed limit, which was the cause of the accident of the young driver.

### 5.3. Relationship between the Behavior in Driving and the Behavior in Preparing to Start the Car

From Table 11 and Table 12, and Figure 9a,b, the difference between the CS group and NCS group in terms of the *θ*^(*y*)^_avg_ of the older driver in Experiment III and IV was statistically significant. From Table 13, there was a strong correlation between *θ*^(*y*)^_avg_ of the older driver in driving and *θ*^(*y*)^_avg_ of the older driver in preparing to start the car. Therefore, the coping skills of the older driver can be predicted from *θ*^(*y*)^_avg_ in preparing to start the car. On other hand, the difference between the CS group and NCS group in terms of the ω^(*x*)^_avg_ of the older driver in Experiment III and IV was not statistically significant. The average of subjective impatience of the drivers in Experiments III and IV was less than 4. Therefore, the older driver felt calm when preparing to start a car. Since the older driver felt calm, the driver’s hip joint did not narrow. Therefore, in ω^(*x*)^_avg_ of the older driver, there is no difference between the CS group and NCS group.

From Table 11 and Table 12, and Figure 9c,d, when the young driver felt calm, the tilt angle of the left foot of the young driver in the NCS group was more perpendicular to the ground than that in the CS group. However, there was no correlation between the young driver’s driving behavior and the young driver’s behavior when preparing to start the car. Besides, it was suggested that the young driver of the NCS group who felt impatient drove a car at a speed exceeding the speed limit, which was the cause of the accident of the young driver. Therefore, the coping skills of the young driver cannot be predicted from *θ*^(*y*)^_avg_ in driving or preparing to start the car.

### 5.4. Prediction Accuracy of Coping Skills by CP System

Table 13 shows that the CP system predicted with 92% accuracy on average based on *θ*^(*y*)^_avg_ and ω^(*x*)^_avg_ of the older driver while driving the car. From Figure 7a, CS and NCS can be classified by setting the threshold value at *θ*^(*y*)^_avg_ = −60 degrees. Therefore, the CP system using random forest was able to predict CS and NCS with high accuracy based on *θ*^(*y*)^_avg_ and ω^(*x*)^_avg_ of an older driver while driving a car. Table 13 shows the CP system predicted with 94% accuracy on average based on *θ*^(*y*)^_avg_ and ω^(*x*)^_avg_ of the older driver while preparing to start the car. From Figure 7b, CS and NCS can be classified by setting the threshold value at *θ*^(*y*)^_avg_ = −60 degrees. Besides, there was a strong correlation between *θ*^(*y*)^_avg_ of the older driver in driving and *θ*^(*y*)^_avg_ of the older driver in preparing to start the car. Therefore, the CP system using random forest was able to predict CS and NCS with high accuracy based on *θ*^(*y*)^_avg_ and ω^(*x*)^_avg_ of an older driver while preparing to start the car. Related works predicted driver risk [33], typical dangerous driving behavior such as chasing a preceding vehicle [34] or driving while operating a mobile phone [35], and illegal drivers [36]. However, there is still not a prediction about the coping skills of older drivers in the face of unexpected situations; although, it is not possible to make a direct comparison because the prediction targets are different, the related study predicts dangerous driving behavior with an accuracy of 92% [34], and the accuracy of this study is about the same even when compared with related work. Therefore, the CP system is effective.

## 6. Conclusions

We proposed a coping skills prediction system. As a result of the experiment, we showed that the CP system predicted coping skills with an accuracy of 92% or 94% on average from the tilt angle and the angular velocity of the left foot while driving or preparing to start the car. Moreover, we clarified that the tilt angle of the left foot of a driver without coping skills is perpendicular to the ground compared to a driver with coping skills. This study is expected to contribute to the prevention of traffic accidents that occur in the face of an unexpected situation.

The CP system is not intended to be used alone. This study predicts coping skills, which are some of the skills of older drivers. When evaluating older drivers, it is necessary to evaluate cognitive skills and driving skills along with coping skills. Therefore, if the user wants to measure coping skills, he/she should wear them on his/her left foot, and if he/she wants to measure other abilities, he/she should measure the maximum value of steering and braking as in the related research. The simulator was built to imitate a real car. However, there is a difference between driving on the simulator and driving the actual vehicle. Therefore, we will investigate the applicability of the CP system in actual vehicles in the future.

## Figures and Tables

**Figure 1 sensors-21-02099-f001:**
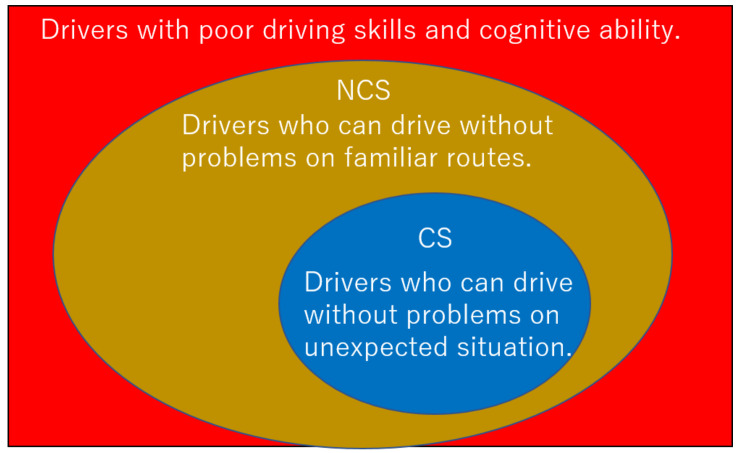
The Venn diagram of those with coping skills (CS) and without coping skills (NCS).

**Figure 2 sensors-21-02099-f002:**
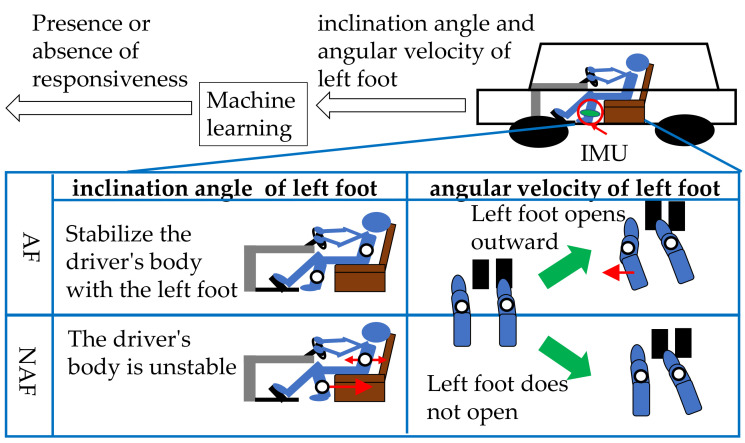
The coping skills prediction (CP) system in which machine learning predicts coping skills from the tilt angle and angular velocity of the left foot when an older driver is driving or starts preparation movement.

**Figure 3 sensors-21-02099-f003:**
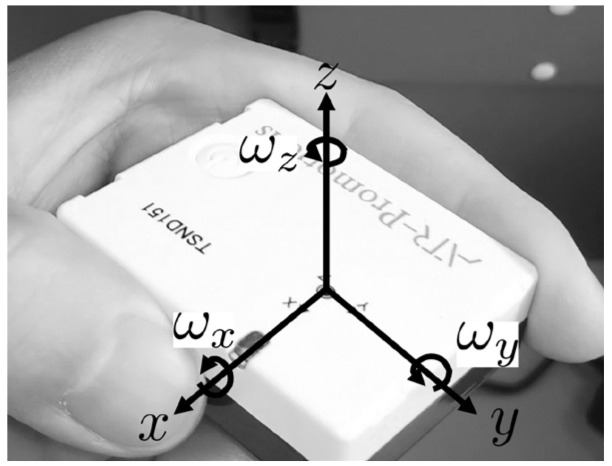
Tilt measurement unit (IMU) coordinate system.

**Figure 4 sensors-21-02099-f004:**
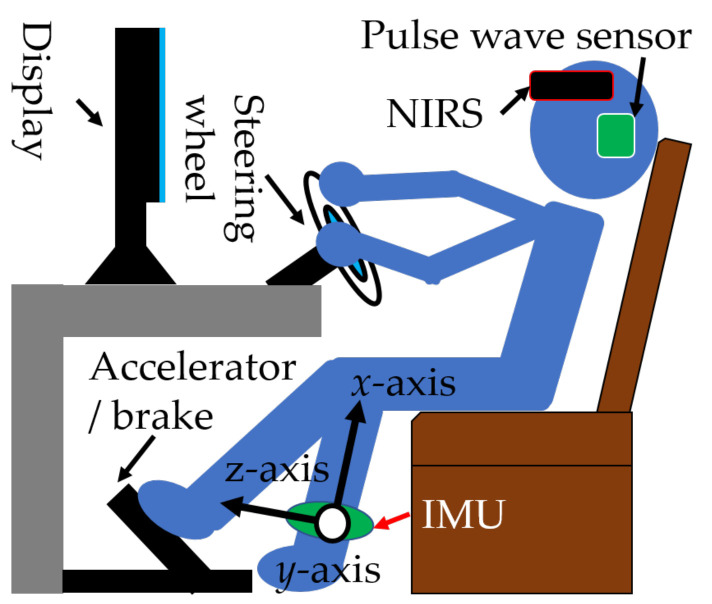
Measurement environment.

**Figure 5 sensors-21-02099-f005:**
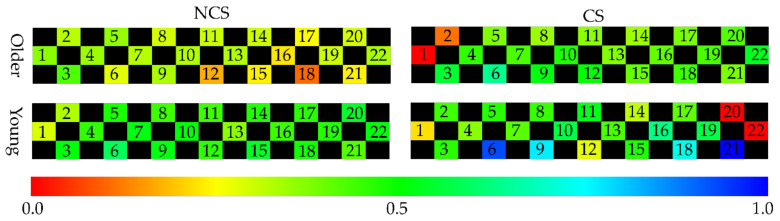
Heat map of brain activity when the driver feels impatient.

**Figure 6 sensors-21-02099-f006:**
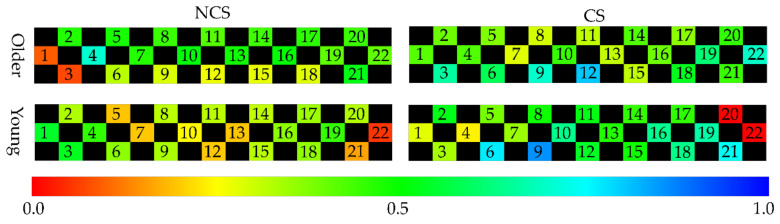
Heat map of brain activity when the driver feels calm.

**Figure 7 sensors-21-02099-f007:**
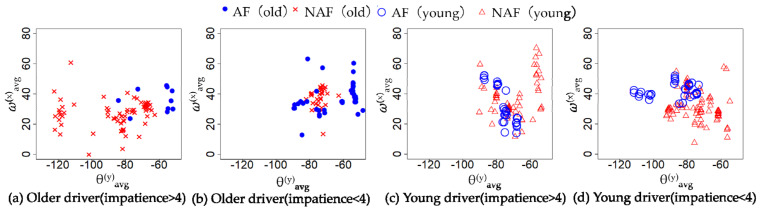
The *y*−axis tilt angle and angular velocity in the *x*−axis when the driver feels impatient or calm.

**Figure 8 sensors-21-02099-f008:**
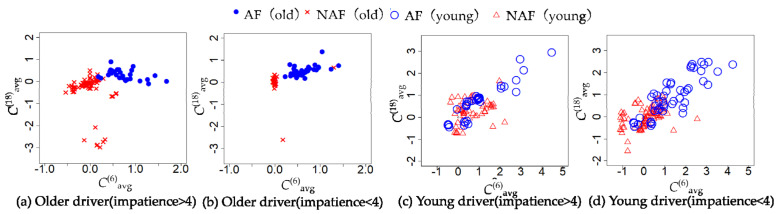
Near−infrared spectroscopy (NIRS) channels 6 and 18 when the driver feels impatient or calm.

**Figure 9 sensors-21-02099-f009:**
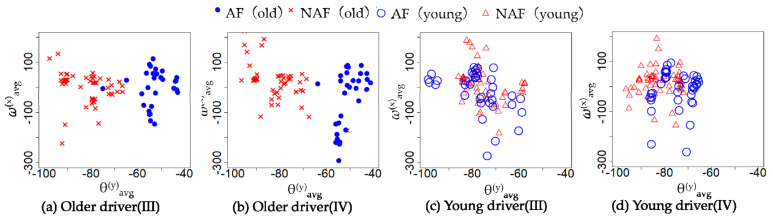
Behavioral characteristics of NCS and CS groups in Experiments III and IV.

**Figure 10 sensors-21-02099-f010:**
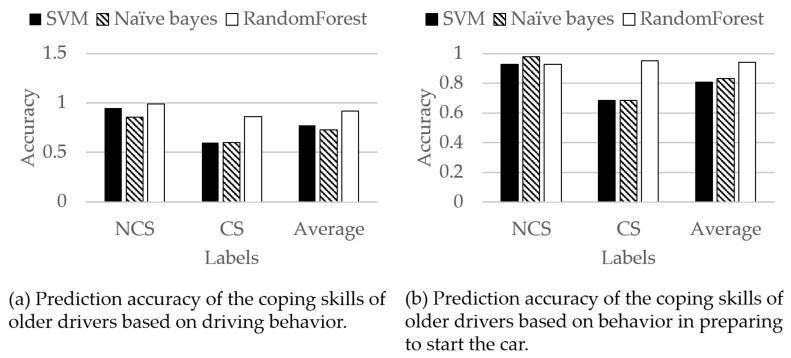
Prediction accuracy of coping skills for older drivers.

**Table 1 sensors-21-02099-t001:** Cases where collision accidents are included in the driving route are likely to occur.

Identifier	Case
1	Overtake parked vehicles in the direction of travel.
2	A child jumps out in front of the vehicle.
3	A motorcycle makes a sudden right turn.
4	Sudden braking of the left-turn vehicle.
5	Change the lane of the vehicle in the direction of travel.
6	Parked vehicle after turning left.
7	Sudden braking of vehicles ahead due to the traffic light switching to red.
8	A child jumps out between parked vehicles.
9	Pass by the side of a vehicle parked in the opposite lane.
10	A motorcycle coming straight from the blind spot of a right-turning vehicle in the opposite lane.
11	A taxi in front suddenly brakes to pick up passengers.
12	Older people crossing a road without traffic lights.
13	The rushing of crossing pedestrians when the traffic signal changes.
14	Pedestrians rushing when the traffic signal changes.
15	A traffic accident was caused by a driver thanking another driver for letting him go first at a junction.

**Table 2 sensors-21-02099-t002:** The specifications of the measuring equipment.

Device	Model Number	Manufacturer	Measurement	Sampling Frequency
IMU	TSND151	ATR-Promotions	Acceleration, angular velocity, and tilt angle of each axis	50 Hz
NIRS	WOT-220	Neu	Cerebral blood flow (22CH)	5 Hz
Pulse wave sensor	Vital Mater	Taos	VLF, LF, HF, TP, HR, RR	1000 Hz
Drive simulator	ACM300	Hitachi KE Systems	The amount of operation of the braking, accelerator, and steering	50 Hz

**Table 3 sensors-21-02099-t003:** Subjective impatience and number of accidents in young and old drivers in Experiment I and Experiment II.

Experiment	Measurement	Avg ± SD		Statistics	*p*-Value
		Young	Old		
I	Impatience	4.2 ± 0.9	4.3 ± 1.8	0.00	1.00
	Accidents	3.4 ± 2.4	4.3 ± 2.3	0.85	0.41
II	Impatience	2.8 ± 1.3	4.1 ± 2.0	1.64	0.13
	Accidents	1.6 ± 0.9	1.6 ± 1.4	−0.16	0.88

**Table 4 sensors-21-02099-t004:** Subjective impatience and number of accidents in young and old drivers in Experiment I and Experiment II.

Set	Measurement	Avg ± SD		Statistics	*p*-Value
		I	II		
∑	Impatience	4.3 ± 1.3	3.4 ± 1.7	−1.64	0.11
	Accidents	3.7 ± 2.3	1.6 ± 1.1	−3.80	0.00

**Table 5 sensors-21-02099-t005:** Three characteristic elements with high statistics.

Set	Characteristic Elements	Avg ± SD		Statistics	*p*-Value
		I	II		
∑	The minimum value of HR	73.7 ± 11.3	69.8 ± 9.49	−4.85	0.00
	Average of HR	80.9 ± 11.5	76.4 ± 9.77	−4.84	0.00
	The maximum value of HR	88.5 ± 12.0	83.6 ± 10.9	−4.60	0.00

**Table 6 sensors-21-02099-t006:** Subjective impatience and number of accidents in CS and NCS groups in Experiment I and II.

Experiment	Measurement	Avg ± SD		Statistics	*p*-Value
		NCS	CS		
I	Impatience	4.4 ± 1.5	4.1 ± 1.1	−0.55	0.59
	Accidents	5.1 ± 1.7	1.4 ± 0.8	-	-
II	Impatience	3.3 ± 1.9	3.4 ± 1.6	0.29	0.78
	Accidents	1.7 ± 1.1	1.6 ± 1.3	−0.38	0.71

**Table 7 sensors-21-02099-t007:** Characteristics of behavior and cognition of older drivers when they feel impatient.

Characteristic Elements	Avg ± SD		Statistics	*p*-Value
	NCS	CS		
Average of steering	−3.89 ± 46.0	0.82 ± 26.6	0.18	0.86
Average of accelerator	21.5 ± 13.3	17.7 ± 8.30	−1.61	0.11
Maximum of braking	65.5 ± 34.9	82.0 ± 30.1	3.47	0.00
Average of tilt angle in the *y*-axis	−86.2 ± 18.7	−51.4 ± 13.8	16.7	0.00
Average of angular velocity in the *x*-axis	27.5 ± 9.14	43.6 ± 11.8	10.6	0.00
Average of cerebral blood flow in channel 6	−0.01 ± 0.23	0.70 ± 0.32	46.4	0.00
Average of cerebral blood flow in channel 18	−0.38 ± 0.84	0.35 ± 0.24	17.7	0.00

**Table 8 sensors-21-02099-t008:** Characteristics of behavior and cognition of older drivers when they feel calm.

Characteristic Elements	Avg ± SD		Statistics	*p*-Value
	NCS	CS		
Average of steering	−14.5 ± 65.2	−0.95 ± 45.2	1.07	0.29
Average of accelerator	18.2 ± 11.0	12.6 ± 7.48	−2.03	0.05
Maximum of braking	10.2 ± 11.1	10.4 ± 15.7	−1.85	0.07
Average of tilt angle in the *y*-axis	−74.2 ± 3.73	−64.4 ± 13.5	3.19	0.00
Average of angular velocity in the *x*-axis	37.7 ± 6.48	38.0 ± 9.73	−0.56	0.58
Average of cerebral blood flow in channel 6	0.05 ± 0.23	0.63 ± 0.25	15.4	0.00
Average of cerebral blood flow in channel 18	−0.01 ± 0.51	0.51 ± 0.21	16.6	0.00

**Table 9 sensors-21-02099-t009:** Characteristics of behavior and cognition of young drivers when they feel impatient.

Characteristic Elements	Avg ± SD		Statistics	*p*-Value
	NCS	CS		
Average of steering	−4.19 ± 43.3	−1.25 ± 42.0	0.85	0.40
Average of accelerator	24.1 ± 9.48	16.9 ± 8.08	−3.97	0.00
Maximum of braking	82.6 ± 21.1	84.6 ± 22.6	0.66	0.51
Average of the tilt angle of the left foot in the *y*-axis	−69.3 ± 10.8	−75.5 ± 6.25	−2.97	0.00
Average of the angular velocity of the left foot in the *x*-axis	38.6 ± 13.0	32.1 ± 12.6	−2.32	0.03
Average of cerebral blood flow in channel 6	0.63 ± 0.66	1.01 ± 1.11	1.78	0.08
Average of cerebral blood flow in channel 18	0.41 ± 0.51	0.73 ± 0.80	2.31	0.02

**Table 10 sensors-21-02099-t010:** Characteristics of behavior and cognition of young drivers when they feel calm.

Characteristic Elements	Avg ± SD		Statistics	*p*-Value
	NCS	CS		
Average of steering	−7.63 ± 40.9	−4.93 ± 37.6	0.41	0.68
Average of accelerator	19.2 ± 8.54	19.9 ± 9.40	0.24	0.81
Maximum of braking	82.5 ± 17.7	74.4 ± 22.5	−2.08	0.04
Average of tilt angle in the *y*-axis	−74.7 ± 10.6	−89.5 ± 13.6	−6.20	0.00
Average of angular velocity in the *x*-axis	33.7 ± 11.4	42.3 ± 13.7	5.77	0.00
Average of cerebral blood flow in channel 6	0.16 ± 0.68	1.35 ± 1.13	7.28	0.00
Average of cerebral blood flow in channel 18	0.15 ± 0.47	0.93 ± 0.91	5.41	0.00

**Table 11 sensors-21-02099-t011:** Behavioral characteristics of NCS and CS groups in Experiment III.

Group	Characteristic Elements	Avg ± SD		Statistics	*p*-Value
		NCS	CS		
Older driver	*θ* ^(*y*)^ _avg_	−81.8 ± 8.47	−52.8 ± 6.60	69.7	0.00
	*w* ^(*x*)^ _avg_	2.14 ± 62.4	−1.77 ± 71.8	0.16	0.88
Young driver	*θ* ^(*y*)^ _avg_	−75.8 ± 7.85	−79.0 ± 13.1	−0.15	0.88
	*w* ^(*x*)^ _avg_	31.9 ± −30.3	−30.3 ± 110.4	−2.06	0.04

**Table 12 sensors-21-02099-t012:** Behavioral characteristics of NCS and CS groups in Experiment IV.

Group	Characteristic elements	Avg ± SD		Statistics	*p*-value
		NCS	CS		
Older driver	*θ* ^(*y*)^ _avg_	−81.6 ± 9.20	−50.7 ± 5.16	44.27	0.00
	*w* ^(*x*)^ _avg_	24.2 ± 73.8	−41.7 ± 115.0	−1.85	0.07
Young driver	*θ* ^(*y*)^ _avg_	−88.9 ± 17.6	−74.8 ± 7.11	6.78	0.00
	*w* ^(*x*)^ _avg_	20.1 ± 69.1	−9.87 ± 76.2	−1.72	0.09

**Table 13 sensors-21-02099-t013:** The correlation coefficients between θ(y)avg and ω(x)avg in Experiment I and II and the left foot movement θ(y)avg and ω(x)avg in the Experiment III and IV.

Group	Characteristic Elements	Correlation Coefficient	Statistics	*p*-Value
Older driver	*w* ^(*x*)^ _avg_	−0.25	−0.62	0.56
	*θ* ^(*y*)^ _avg_	0.84	3.72	0.01
Young driver	*w* ^(*x*)^ _avg_	0.34	1.04	0.33
	*θ* ^(*y*)^ _avg_	0.48	1.54	0.16

**Table 14 sensors-21-02099-t014:** Results of predicting the coping skills of older drivers based on driving behavior using random forest.

		Labels		Accuracy
		NCS	CS	
prediction results	NCS	68	4	0.99
	*CS*	1	25	0.86

**Table 15 sensors-21-02099-t015:** Results of predicting the coping skills of older drivers based on behavior in preparing to start the car using random forest.

		Labels		Accuracy
		NCS	CS	
prediction results	NCS	91	3	0.93
	*CS*	7	57	0.95

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
