# Peer review of "Predicting the Coping Skills of Older Drivers in the Face of Unexpected Situation"

_sensors, 2021, doi:10.3390/s21062099_

Round 1

Reviewer 1 Report

The following are my comments:

  1. There are some grammar mistakes, e.g., line 9 “…may causes…” should be “… may cause …”, line 22 “…, hey often …” should be “…, they often …”, and etc.
  2. What is “coping skills” and “unexpected situations”? It is suggested to be defined or explained in detail. These are very broad concepts and I believe that only the tilt angle of the left foot is not enough to include all information required to measure all coping skills and deal with all unexpected situations.
  3. As mentioned in the introduction, older drivers response slower and so why not evaluate the coping skill by response time or similar indexes, which can be measured by many proven, easy and effective methods? There many What is the advantage of the proposed method? easy application? Low cost? More accurate or others?
  4. The author should give out the fundamental, principle, consideration, assumption and etc. about the proposed measurement index about coping skills. Not just give out it directly and using some intelligent algorithms to learn the model parameters. Theoretically, a fine fitting result can always be achieved by some model structure when the dataset is given.

Author Response

Response to Reviewer 1 Comments

Thank you for your review. We have carefully corrected this paper.

Point 1: There are some grammar mistakes, e.g., line 9 “…may causes…” should be “… may cause …”, line 22 “…, hey often …” should be “…, they often …”, and etc.

Response 1: We apologize for any typographical errors. We have corrected the typographical error in the paper(line 9 and line22).

Point 2: What is “coping skills” and “unexpected situations”? It is suggested to be defined or explained in detail. These are very broad concepts and I believe that only the tilt angle of the left foot is not enough to include all information required to measure all coping skills and deal with all unexpected situations.

Response 2: The older drivers are prone to dynamic hazard accidents such as collisions with other vehicles [15]. The driver's physical function is impaired by mental stress [16]. We define coping skills as skills to properly step the brake and stop the car when faced with an unexpected dynamic hazard under a mental workload such as impatience or nervousness.

As you indicated, keep in mind that the CP system predicts some of the capabilities re-quired to drive. Therefore, it is necessary to comprehensively evaluate the driving skills of the driver by using a method such as DRT and a CP system together (line 83-87 and line 534-547).

Point 3: As mentioned in the introduction, older drivers response slower and so why not evaluate the coping skill by response time or similar indexes, which can be measured by many proven, easy and effective methods? There many What is the advantage of the proposed method? easy application? Low cost? More accurate or others?

Response 3: DRT indirectly measures cognitive load by asking the driver to respond when they detect a small light in their peripheral vision. DRT is an international standard, which can be safely applied with no appreciable effect on driving performance. However, in this study, we did not use DRT because we used NIRS to measure cognitive load directly.

CP system predicted coping skills to properly step the brake and stop the car when faced with an unexpected dynamic hazard under a mental workload such as impatience or nervousness. DRT indirectly measures cognitive load. The measurement target is different between the CP system and DRT. The CP system has the following ad\vantages:

  1. A) No special equipment is required. The tilt angle of the left foot can be ob-tained from the in-vehicle camera.
  2. B) The CP system is inexpensive.
  3. C) CP system can predict coping skills to properly step the brake and stop the car when faced with an unexpected dynamic hazard under a mental workload.

However, keep in mind that the CP system predicts some of the capabilities re-quired to drive. Therefore, it is necessary to comprehensively evaluate the driving skills of the driver by using a method such as DRT and a CP system together (line 256-262 and line 534-547)

Point 4: The author should give out the fundamental, principle, consideration, assumption and etc. about the proposed measurement index about coping skills. Not just give out it directly and using some intelligent algorithms to learn the model parameters. Theoretically, a fine fitting result can always be achieved by some model structure when the dataset is given.

Response 4: Added Chapter 2.2 to explain the Fundamental principle. The driver's movements when driving a car are different from the driver's movements when preparing to start the car. However, these movements have in common that the driver recognizes, judges, and controls the movement within the time limit. Therefore, a correlation is expected between the driver's movements during preparation for starting the car and the driver's movements during driving the car. The amount of braking can be obtained directly from the car control device. The amount of braking can be obtained indirectly from the movement of the right foot. However, it is difficult to measure the stability of the body and muscle tension. There-fore, in this study, we focused on the tilt angle and angular velocity of the left foot.

 We have added typical machine learning predictions. We also added a description of machine learning parameters(line 104, line113-117, line 130-133, line182-186, line 480-483, line 494-497,Figure 10)

Reviewer 2 Report

I think the paper presents an interesting topic. Also the revised version is much improved. Hence, I recommend the paper for publication.

Author Response

Response to Reviewer 2 Comments

Thank you for your review.

Reviewer 3 Report

The paper was improved according to the comments on the previous version. Only one comment, the points I, II, III, and IV must be better contextualized.

Author Response

Thank you for your review. We have carefully corrected this paper.

Point 1: The paper was improved according to the comments on the previous version. Only one comment, the points I, II, III, and IV must be better contextualized.

Response 1: This point I, II, III, IV was modified in consideration of the possibility that it indicates the experiment I, II, III, IV. We have also responded to points 1, 2, 3 and 4 of the previous review. In Experiment I, the 15 cases in Table 1 are unexpected situations since the driver drives on an unfamiliar road. In Experiment I, the driver feels impatient and nervous because the driver must drive the route within the time limit. We analysed the correlation between the left foot movement in driving (Experiment I and II) and the left foot movement during preparing to start the car (Experiment I and II). By analysing the correlation, we show that the coping skills can be predicted from the movements of the left foot of the older driver during preparing to start the car. Subject information was added to abstract and Introduction. Also added an explanation of the basic principle. We have also added a new discussion of comparison between DRT and this study(line 12-24, line 59-67, line73-76, line83-87, line 113-117, line130-133, line182-186, line 214-217, line236-240, line 256-262, line 480-483, line 494-497, figure 10, and line 534-547).

Round 2

Reviewer 1 Report

All comments have been well addressed by the auhtor.